# Multi-ethnic transcriptome-wide association study of prostate cancer

Peter N. Fiorica[1,2], Ryan Schubert[2,3,4], John D. Morris[2,3], Mohammed Abdul Sami[2], Heather E. Wheeler [1,2,3,5]*

**1** Department of Chemistry & Biochemistry, Loyola University Chicago, Chicago, IL, United States of America, **2** Department of Biology, Loyola University Chicago, Chicago, IL, United States of America, **3** Program in Bioinformatics, Loyola University Chicago, Chicago, IL, United States of America, **4** Department of Statistics, Loyola University Chicago, Chicago, IL, United States of America, **5** Department of Public Health, Loyola University Chicago, Chicago, IL, United States of America

* hwheeler1@luc.edu

**Data Availability Statement:** Phenotype and genotype data are available from the NCBI database of Genotypes and Phenotypes (dbGaP) accession number phs000306.v4.p1. All scripts

## Abstract

The genetic risk for prostate cancer has been governed by a few rare variants with high penetrance and over 150 commonly occurring variants with lower impact on risk; however, most of these variants have been identified in studies containing exclusively European individuals. People of non-European ancestries make up less than 15% of prostate cancer GWAS subjects. Across the globe, incidence of prostate cancer varies with population due to environmental and genetic factors. The discrepancy between disease incidence and representation in genetics highlights the need for more studies of the genetic risk for prostate cancer across diverse populations. To better understand the genetic risk for prostate cancer across diverse populations, we performed PrediXcan and GWAS in a case-control study of 4,769 self-identified African American (2,463 cases and 2,306 controls), 2,199 Japanese American (1,106 cases and 1,093 controls), and 2,147 Latin American (1,081 cases and 1,066 controls) individuals from the Multiethnic Genome-wide Scan of Prostate Cancer. We used prediction models from 46 tissues in GTEx version 8 and five models from monocyte transcriptomes in the Multi-Ethnic Study of Atherosclerosis. Across the three populations, we predicted 19 gene-tissue pairs, including five unique genes, to be significantly (*lfsr* < 0.05) associated with prostate cancer. One of these genes, *NKX3-1*, replicated in a larger European study. At the SNP level, 110 SNPs met genome-wide significance in the African American study while 123 SNPs met significance in the Japanese American study. Fine mapping revealed three significant independent loci in the African American study and two significant independent loci in the Japanese American study. These identified loci confirm findings from previous GWAS of prostate cancer in diverse populations while PrediXcan-identified genes suggest potential new directions for prostate cancer research in populations across the globe.

## Introduction

In the past two decades, genome-wide association studies (GWAS) have evolved as a critical method to detect and characterize genomic loci affecting susceptibility to various polygenic

and notes for this study can be found at https://github.com/WheelerLab/Prostate-Cancer-Study. Summary statistics are available in the GWAS Catalog under accessions GCST90002423, GCST90002424, and GCST90002425.

**Funding:** This work is supported by the NIH National Human Genome Research Institute Academic Research Enhancement Award R15 HG009569 (HEW), the Loyola Carbon Undergraduate Research Fellowship (PNF), the Loyola Mulcahy Scholarship (PNF, JDM, MAS), and the Loyola Provost Fellowship (JDM). Funding support for the GENEVA Prostate Cancer study was provided through the National Cancer Institute (R37CA54281, R01CA6364, P01CA33619, U01CA136792, and U01CA98758) and the National Human Genome Research Institute (U01HG004726). Assistance with phenotype harmonization, SNP selection, data cleaning, meta-analyses, data management and dissemination, and general study coordination, was provided by the GENEVA Coordinating Center (U01HG004789-01). The content is solely the responsibility of the authors and does not necessarily represent the official views of the National Institutes of Health.

**Competing interests:** The authors have declared that no competing interests exist.

disorders. As this important method to detect genetic associations has grown, individuals of non-European ancestries have made up less than 22% of all GWAS. Individuals of African, East Asian, and Latin American ancestries made up 2.03%, 8.21%, and 1.13%, respectively [1]. These populations are similarly poorly represented in GWAS of prostate cancer. Individuals of European ancestries make up over 85% of discovery GWAS, while individuals of African, East Asian, and Latin American ancestries make up less than 11%, 3%, and 1%, respectively [2]. Individuals of African American or Latin American ancestries are far more diverse due to more recent genetic admixture compared to individuals with East Asian ancestries [3]. This means that individuals who identify as Latin American or African American may have genomes composed of linkage disequilibrium (LD) blocks from Africa, Europe, and the Americas. Admixture and difference in population structure can lead to complexities in studying genetic associations within diverse populations. Nonetheless, these populations need to be studied to better understand disease risk across the globe. The lack of representation of non-European populations in GWAS can lead to further health disparities from non-transferable findings across populations. Since allele and haplotype frequencies differ across populations, susceptibility to disease will vary with these frequencies [4]. Elucidating this susceptibility has grown increasingly important since many disease risk prediction models lose performance accuracy across populations [5]. Having accurate models to predict disease risk is especially important for prostate cancer since disease risk is noticeably different across populations.

Prostate Cancer is the most commonly diagnosed cancer and second leading cause of cancer death among African American men; one in seven black men will be diagnosed with prostate cancer in his lifetime compared to one in nine white men [6]. Risk factors for prostate cancer include age, family history of disease, and African ancestries. The genetic component of prostate cancer is made of rare variants with high penetrance and many common variants with lower penetrance [7, 8]. Adding to the complexity of this analysis, prostate cancer is a remarkably heterogeneous phenotype with various molecular and physical classifications [9]. While prostate cancer susceptibility is increased in African Americans, prostate cancer risk is poorly understood in individuals of East Asian ancestries. The age adjusted incidence rate of prostate cancer in East Asian Americans is nearly three times that of native East Asian individuals [10]. Better understanding of how alleles specific to East Asian populations influence prostate cancer risk could reveal new underlying disease biology. Latin American individuals are the least studied of the three populations. The lack of representation could be due to low rates of incidence of prostate cancer in Central and South America; however, Latin American countries are estimated to have the second highest increase in risk for the disease by 2040 [11].

Of the 67 published prostate cancer GWAS in the National Center for Biotechnology Information (NCBI) GWAS Catalog, only 16 studies included individuals of non-European ancestries [12]. These GWAS have identified 8q24 as a region carrying susceptibility loci for prostate cancer. While some fine-mapping of this region has been done in diverse populations, transcriptome-wide association studies (TWAS) of prostate cancer have not been conducted across diverse populations [13, 14]. TWAS serves as a systematic method for integrating expression quantitative trait loci (eQTL) data from GWAS [15–17]. Since TWAS uses gene expression as an intermediate phenotype, it has a functional advantage over GWAS. One of the largest GWAS published up to this point that included over 140,000 individuals of European ancestries was the Prostate Cancer Association Group to investigate Cancer-Associated Alterations in the Genome (PRACTICAL) Consortium [8]. Two of the largest gene-based association studies specific to prostate cancer were TWAS of the PRACTICAL GWAS summary statistics [18, 19]. While these studies provided insight into genes associated with prostate cancer in European subjects, they provided little insight into genes affected by ancestry-specific SNPs in diverse populations. Moreover, broader and more accurate genotype and

gene expression imputation panels have been created since the studies in diverse populations were published.

We seek to better understand the genetic architecture of gene expression for prostate cancer in African American, Japanese American, and Latin American populations. To do this, we performed a standard case-control GWAS across 4,769 African American subjects, 2,147 Latin American subjects, and 2,199 Japanese American subjects. We used a deterministic approximation of posteriors for GWAS (DAP-G) to fine-map the 8q24 susceptibility region [20]. In addition to GWAS, we performed TWAS using PrediXcan across 46 tissues from GTEx version 8 and five models from the Multi-Ethnic Study of Atherosclerosis. We replicated our data in an S-PrediXcan application to the PRACTICAL summary statistics [15, 21]. All scripts and notes for this study can be found at https://github.com/WheelerLab/Prostate-Cancer-Study.

## Methods

### Genotype and phenotype data

Phenotype and genotype data for all individuals in this study were downloaded from the NCBI database of Genotypes and Phenotypes (dbGaP) accession number phs000306.v4.p1. Our project was determined exempt from human subjects federal regulations under exemption number 4 by the Loyola University Chicago Institutional Review Board (project number 2014). Participants were mainly self-reported ethnicities of African Americans, Japanese Americans, or Latin Americans. Cases of cancer within these men were identified by annual linkage to the National Cancer Institute (NCI) Surveillance, Epidemiology, and End Results (SEER) registries in California and Hawaii. Whole genome genotyping was performed on Illumnia Human 1M-Duov3_B and Human660W-Quad_v1_A array platforms surveying 1,185,051 and 592,839 single nucleotide polymorphisms (SNP), respectively [22, 23]. The final association analyses included 4,769 African American subjects (2,463 cases and 2,306 controls), 2,147 Japanese American subjects (1106 cases and 1093 controls), and 2,199 Latin American subjects (1081 cases and 1066 controls) (Table 1).

### Quality control and imputation

After we downloaded the data from dbGaP, we divided the subjects into three groups of their self-reported ethnicities. Standard genome-wide quality control was performed separately on each of the three groups using PLINK [24]. In each group, we removed SNPs with genotyping rates < 99%. We then removed SNPs significantly outside of Hardy-Weinberg Equilibrium ($P < 1 \times 10^{-6}$). We also filtered out individuals with excess heterozygosity, removing individuals at least three standard deviations from mean heterozygosity. We then used PLINK to calculate the first ten principal components of each study when merged with three populations

**Table 1. Population characteristics: Genotype and phenotype data from the three populations went through standard genome-wide quality control and genotype imputation.**

| Population | African American | Japanese American | Latin American |
|---|---|---|---|
| Pre-QC Individuals | 4874 | 2199 | 2147 |
| Post-QC Individuals | 4769 | 2199 | 2147 |
| Cases | 2463 | 1106 | 1081 |
| Controls | 2306 | 1093 | 1066 |
| Pre-QC SNPs | 1,199,187 | 657,366 | 657,366 |
| Post-QC SNPs | 1,077,583 | 540,326 | 539,366 |
| Post-Imputation SNPs | 15,394,464 | 4,623,264 | 7,010,834 |

from HapMap phase 3 [25]. The first three principal components of each group were used to confirm self-identified ethnicity (S1 Fig).

Following the confirmation of ethnicity, filtered genotypes were imputed using the University of Michigan Imputation Server [26]. All three groups of genotypes were imputed separately using minimac4 and Eagle version 2.3 for phasing. For the Japanese American population, 1000 Genomes Phase 3 version 5 (1000G) was used as a reference panel [27]. Both the African American and Latin American groups were imputed with 1000G and the Consortium on Asthma among African-ancestry Populations in the Americas (CAAPA) [28]. Genotypes for all three populations were filtered for imputation accuracy and minor allele frequency (MAF). SNPs with $r^2 < 0.8$ and MAF < 0.01 were removed from the analysis. The union of genotypes imputed with 1000G and CAAPA in African American and Latin American populations were used for downstream analysis. $r^2$ and MAF filtering took place before the merging of genotypes. For SNPs in the intersection of the two imputation panels, the SNP imputed with CAAPA was selected for the GWAS and PrediXcan analysis due to LD similarity between the CAAPA reference panel and the populations studied. The number of SNPs and individuals in each ethnicity at various steps throughout quality control and imputation are reported in Table 1.

## Genome-wide association study

We performed a traditional case-control GWAS of prostate cancer using a logistic regression in PLINK. The first ten principal components were used as covariates to account for global population structure. $P < 5 \times 10^{-8}$ was used as the P-value threshold to denote genome-wide significance. Independent loci were determined and analyzed using deterministic approximation of posteriors for GWAS (DAP-G), an integrative joint enrichment analysis of multiple causal variants [20]. SNPs were clustered into groups with both a cluster posterior inclusion probability (PIP) and an individual SNP PIP. SNPs in the same cluster were identified as linked, and each group of SNPs was considered a locus independent of others. Pairwise LD calculations of $r^2$ were calculated in PLINK [24].

## PrediXcan and S-PrediXcan

We used the gene expression imputation method, PrediXcan, to predict genetically regulated levels of expression for each individual across the three populations [15]. We predicted expression using five models built from monocyte transcriptomes of self-identified European (CAU), Hispanic (HIS), African American (AFA) individuals in the Multi-Ethnic Study of Atherosclerosis (MESA). The two other MESA models were built from combined African American-Hispanic (AFHI) data, and all three populations (ALL) [29]. Additionally, we also predicted expression using the 46 multivariate adaptive shrinkage (mashr) models built from 46 tissues in the Gene-Tissue Expression Project (GTEx) version 8 [30, 31]. Ovary, uterus, and vagina were excluded from the total tissues. In the GTEx version 8 prediction models, only tissues from individuals with European ancestries were used. All gene expression prediction models may be found at http://predictdb.org/. We tested the predicted expression levels for association with the case-control status of individuals in each ethnic population using the first ten principal components as covariates. Significant gene-tissue association were determined after performing an adaptive shrinkage using the R package ashr to account for multiple testing [32]. The adaptive shrinkage calculated a local false sign rate (*lfsr*) for each test. Gene-tissue pairs with *lfsr* < 0.05 were considered significant. We chose *lfsr* over traditional false discovery rate because *lfsr* accounts for both effect size and standard error for each gene-tissue pair. To confirm significant gene-tissue findings, we used COLOC to investigate colocalization

between the GWAS and eQTLs [33]. We followed up these finding using the summary statistics version of PrediXcan, S-PrediXcan [21]. We applied the same GTEx prediction models to the PRACTICAL summary statistics, which included over 20.3M SNPs [8].

## Results

### Genome-wide association studies

To better characterize the genetic architecture underlying prostate-cancer across non-European populations, we performed a case-control GWAS of prostate cancer across 15M SNPs in nearly 5,000 self-identified African American individuals. Additionally, we performed GWAS across 4.6M SNPs in a nearly 2,200 Japanese American individuals and 7.0M SNPs in 2,147 Latin American individuals. Notably, this GWAS includes imputed SNPs from 1000G and CAAPA, two reference panels not available at the time of the original genotyping and GWAS of this case-control study [22, 23, 27, 34].

In our GWAS of 4,769 self-identified African American individuals, we found 110 SNPs to be significantly associated ($P < 5 \times 10^{-8}$) with prostate cancer. Of these 110 SNPs, 108 of them were located at a previously identified region of chromosome 8 (Fig 1A). Of the SNPs on chromosome 8, rs76595456 was the most significantly associated at $P = 1.01 \times 10^{-15}$. The minor allele (T) of rs76595456 is a SNP found only in individuals of recent African ancestries (Fig 2). Four independent clusters were identified by DAP-G on chromosome 8 at 8q24 (Fig 1C). Of the four clusters, two contained SNPs meeting genome-wide significance, one led by rs76595456 (PIP = 0.942) and the other led by rs72725879 (PIP = 0.994) (S1 Table). rs72725879 was also significant in the Japanese American GWAS (Fig 1B and 1D). Two of 110 SNPs (rs10149068 & rs8017723) were found at a novel locus on chromosome 14. These two SNPs on chromosome 14 are linked ($r^2 = 0.989$) and identified as one locus.

The GWAS of 2,199 Japanese American individuals identified 123 SNPs as genome-wide significant. Every genome-wide significant SNP was located at the 8q24 region of chromosome 8 (Fig 1B). rs1456315 was the most significantly associated SNP in this study ($P = 1.40 \times 10^{-13}$). We identified six distinct clusters of SNPs with DAP-G (Fig 1D). Two of the six clusters contained SNPs meeting genome-wide significance with PIP > 0.5. rs1456315 had not only the lowest P-value, but also the highest PIP (PIP = 0.990) in the Japanese American GWAS (Fig 2). rs1456315 was found to be marginally significant in the African American GWAS ($P = 1.29 \times 10^{-7}$). rs1456315 and rs72725879 are linked ($r^2 = 0.815$) in the Japanese American GWAS population and have similar allele frequencies across East Asian populations while rs1456315 and rs72725879 are less linked ($r^2 = 0.448$) in the African American GWAS population and have divergent allele frequencies across African populations (Fig 2). $r^2$ values between SNPs were calculated using PLINK in each individual case-control study. $D'$ values were calculated using the 1000G African reference population (AFR) in LDlink [27, 37]. The GWAS of 7M SNPs in 2,147 Latin American individuals identified no genome-wide significant SNPs. chr13:106685795 was the most associated SNP ($P = 4.41 \times 10^{-7}$), located on chromosome 13 (S2 Fig).

### Gene-based association studies

We used the gene expression imputation tool, PrediXcan to predict gene expression from genotypes using models built from transcriptomes of 46 tissues in version 8 of GTEx [17, 30]. We also predicted gene expression using five models built from monocyte transcriptome of diverse populations in MESA [29]. After predicting the genetically regulated level of expression for each gene using each of these models, we tested the predicted gene expression for

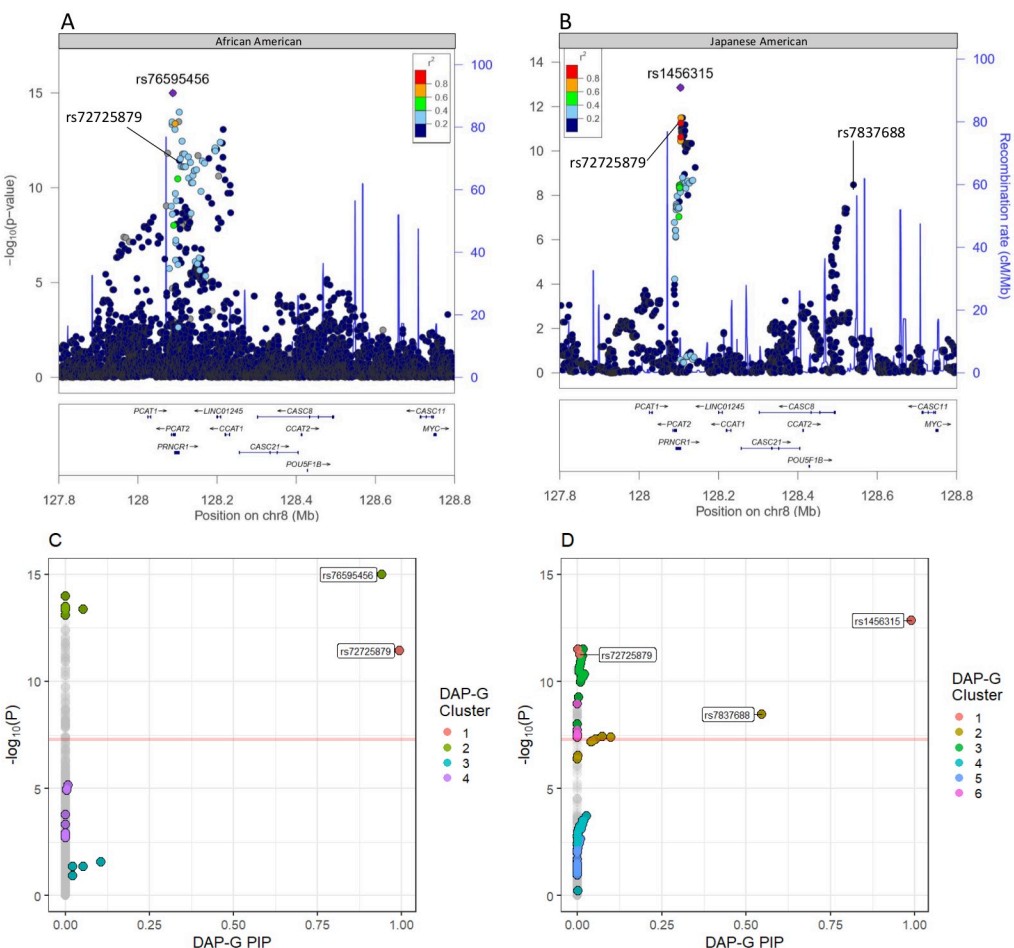

**Fig 1. Fine-mapping of the top prostate cancer GWAS signals in African Americans and Japanese Americans.** (A & B) depict a LocusZoom plots of GWAS results from African American and Japanese American populations, respectively [35]. The most significant SNPs in both GWAS were in the same chromosome 8 region. (A) is plotted using 1000G AFR 2014 LD, and (B) is plotted using 1000G ASN 2014 LD. The y-axis is the $-log_{10}$(P) while the x-axis is location on chromosome 8 measured in megabases (Mb). Color represents the LD $r^2$. (C & D) depict the GWAS $-log_{10}$(P) compared to DAP-G SNP posterior inclusion probabilities (PIP) for the African American and Japanese American populations, respectively [20]. Each point on the plot represents one SNP in each GWAS. The color of each point represents the independent cluster the SNP was assigned to in its respective population. Grey points represent those that were not clustered by DAP-G.

association with the phenotype status of each subject using the first ten principal components as covariates.

In our application of PrediXcan to the 4,769 African American individuals, we predicted expression of 489,459 gene-tissue pairs. We identified two gene-tissue pairs with *lfsr* < 0.05 and nine gene-tissue pairs with *lfsr* < 0.10 across all GTEx version 8 *mashr* prediction models (Table 2; Fig 3). *EBPL* was the gene in all nine of the gene-tissue pairs. The two most significantly associated gene-tissue pairs by *lfsr* were found in cerebellar hemisphere (*lfsr* = 0.0423) and cervical spinal cord (*lfsr* = 0.0485). The gene-tissue pair with the lowest P-value was *KLK3*, a gene encoding for prostate-specific antigen, in Tibial Artery ($P = 3.31 \times 10^{-6}$), but the *lfsr* was not significant (*lfsr* = 0.921). No gene-tissue pairs in the MESA prediction models significantly associated with prostate cancer in the African American population (S3 Fig).

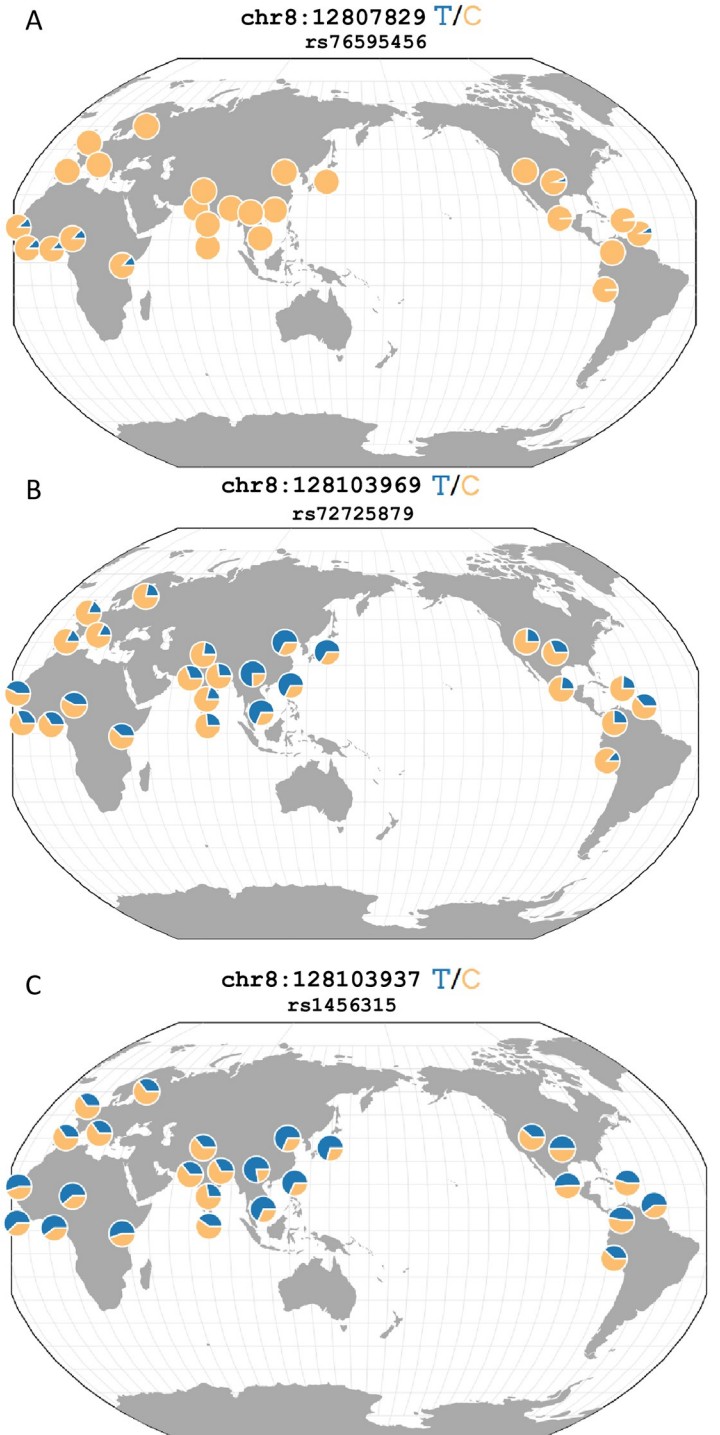

**Fig 2. Global allele frequencies of SNPS significantly associates with prostate cancer.** A depiction of the global minor allele frequencies rs76595456 (A), rs72725879 (B), and rs1456315 (C). (A) rs76595456 represents the most significantly associated SNP in the African American GWAS. The minor allele, T, is found only in populations of recent African ancestries. (B) rs72725879 represents the only SNP to be identified by DAP-G in a cluster in both the African American and Japanese American GWAS. (C) rs1456315 represents the most significantly associated SNP in the Japanese American GWAS. rs1456315 (C) is found in strong LD with rs72725879 (B) ($r^2$ = 0.815) in the Japanese American GWAS population. rs1456315 and rs72725879 are not linked in the African American GWAS population ($r^2$ = 0.448). This figure was adapted from one generated using the Geography of Genetic Variants Browser [36]. SNP position on chromosome 8 is labeled using hg19 coordinates from 1000G.

**Table 2. PrediXcan significant genes significant (*lfsr* < 0.05) gene-tissue pairs from PrediXcan analysis.** *lfsr* represents the local false sign rate as calculated using adaptive shrinkage [32]. P(PRACTICAL) represents P-value for the gene-tissue pair in the S-PrediXcan analysis of the PRACTICAL summary statistics. Beta(PRACTICAL) represents the effect direction and size predicted from S-PrediXcan. "NA" means that the gene was not tested in the PRACTICAL summary statistics.

| Population | Gene | Tissue | lfsr | P | Beta | P (PRACTICAL) | Beta (PRACTICAL) |
|---|---|---|---|---|---|---|---|
| African American | *EBPL* | Brain_Cerebellar_Hemisphere | 0.0423 | 5.25E-05 | -0.022 | 0.3 | 0.006 |
| African American | *EBPL* | Brain_Spinal_cord_cervical_c-1 | 0.0485 | 4.94E-05 | -0.032 | 0.551 | 0.005 |
| Japanese American | *PLCL1* | Adrenal_Gland | 0.0448 | 9.72E-07 | 0.752 | 0.419 | -0.068 |
| Japanese American | *PLCL1* | Artery_Aorta | 0.0435 | 9.72E-07 | 0.738 | 0.419 | -0.067 |
| Japanese American | *NKX3-1* | Brain_Caudate_basal_ganglia | 0.0472 | 1.05E-05 | -0.208 | 3.61E-25 | -0.288 |
| Japanese American | *FAM227A* | Brain_Nucleus_accumbens_basal_ganglia | 0.042 | 9.13E-06 | -0.211 | NA | NA |
| Japanese American | *NKX3-1* | Brain_Putamen_basal_ganglia | 0.0475 | 1.05E-05 | -0.212 | 1.70E-46 | -0.215 |
| Japanese American | *FAM227A* | Brain_Putamen_basal_ganglia | 0.0379 | 8.21E-06 | -0.183 | NA | NA |
| Japanese American | *NKX3-1* | Brain_Substantia_nigra | 0.0209 | 3.80E-06 | -0.238 | 1.18E-20 | -0.288 |
| Japanese American | *FAM227A* | Esophagus_Mucosa | 0.0444 | 9.80E-06 | -0.206 | NA | NA |
| Japanese American | *NKX3-1* | Heart_Left_Ventricle | 0.0469 | 1.05E-05 | -0.202 | 3.61E-25 | -0.28 |
| Japanese American | *FAM227A* | Heart_Left_Ventricle | 0.0434 | 9.22E-06 | -0.222 | NA | NA |
| Japanese American | *NKX3-1* | Liver | 0.0468 | 1.05E-05 | -0.198 | 3.61E-25 | -0.274 |
| Japanese American | *PLCL1* | Lung | 0.0485 | 9.72E-07 | 0.797 | 0.419 | -0.072 |
| Japanese American | *NKX3-1* | Muscle_Skeletal | 0.0413 | 8.02E-06 | -0.147 | 3.61E-25 | -0.375 |
| Japanese American | *PLCL1* | Nerve_Tibial | 0.031 | 1.40E-06 | 0.44 | 0.537 | -0.033 |
| Japanese American | *FAM227A* | Pituitary | 0.0418 | 9.09E-06 | -0.178 | NA | NA |
| Japanese American | *FAM227A* | Prostate | 0.045 | 1.00E-05 | -0.186 | NA | NA |
| Japanese American | *COQ10B* | Thyroid | 0.0418 | 2.29E-07 | -1.138 | 0.298 | 0.13 |

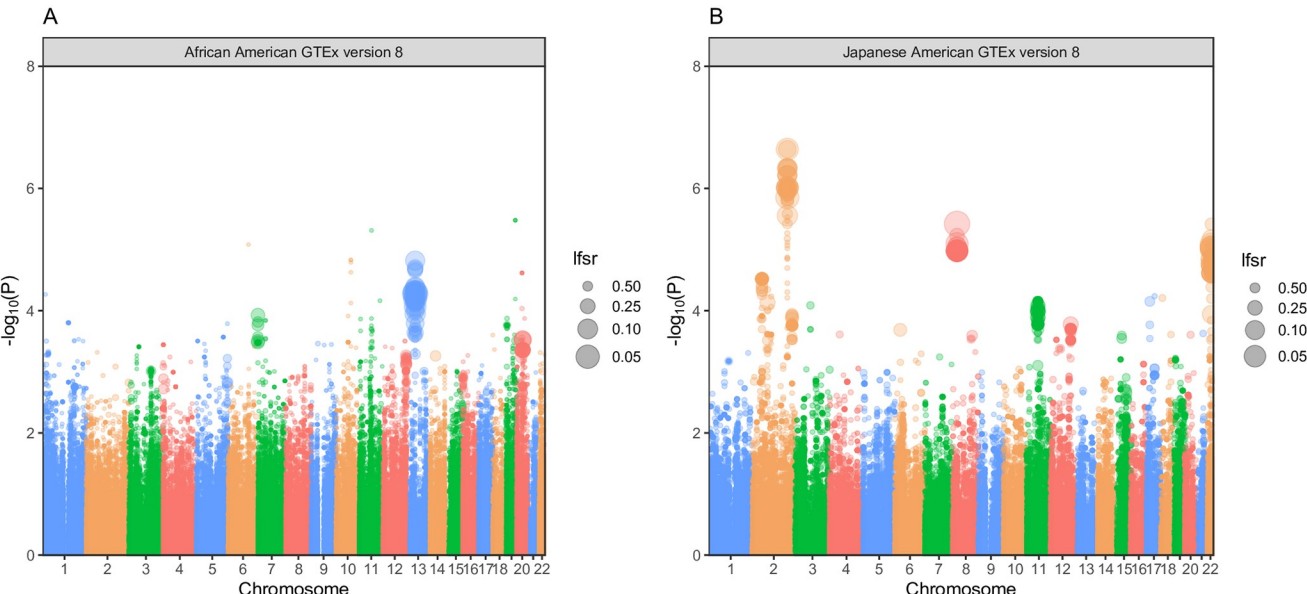

**Fig 3. Prostate cancer PrediXcan results for GTEx predicted genes in African American and Japanese American populations.** (A & B) are Manhattan plots of the PrediXcan results using GTEx version 8 *mashr* gene expression prediction models for the respective African American and Japanese American populations. Each point represents a gene-tissue test for association with prostate cancer via PrediXcan. The y-axis represents the $-log_{10}(P)$ of the gene-tissue test, and the x-axis plots chromosome number. The size of the dot is inversely proportional to its *lfsr*.

We used PrediXcan to impute and associate gene expression with prostate cancer across 270,813 gene-tissue pairs in GTEx version 8 from 2,199 Japanese American individuals. We found seventeen gene-tissue pairs to be significantly associated with prostate cancer in this population (Table 2). Of these seventeen gene-tissue pairs, four unique genes were identified: *PLCL1*, *NKX3-1*, *FAM227A*, *COQ10B*. The most significantly associated gene-tissue pair by P-value was *COQ10B* in Thyroid ($P = 2.28 \times 10^{-7}$). When we applied PrediXcan to the Latin American study, we predicted expression of 411,366 gene-tissue pairs in GTEx version 8. No genes were significantly associated with prostate cancer by *lfsr* or P-value (S4 Fig).

We attempted to replicate our PrediXcan findings by applying S-PrediXcan to GWAS summary statistics of the PRACTICAL meta-analysis of over 140,000 individuals of European ancestries performed by Schumacher et al. [8]. 424,518 gene-tissue pairs were tested from the summary statistics using GTEx version 8 prediction models. When we compared these summary level results to our findings, only *NKX3-1* replicated with a P-value meeting Bonferroni significance ($P < 1.178 \times 10^{-7}$). Similar to the findings of our TWAS, the direction of effect predicted by S-PrediXcan was negative, associating decreased expression of *NKX3-1* with prostate cancer (Table 2). Not enough SNPs were not present in the *FAM227A* prediction model to reliably predict expression from PRACTICAL summary statistics.

## Discussion

Broadly, the findings of this study confirm previously well-established information about the genetics of prostate cancer in diverse populations; however, our study differs from previous ones since we performed the first TWAS of prostate cancer in these populations using mashr prediction models from GTEx version 8 and diverse prediction models from MESA [17, 29]. Two of the main findings of this study were identifying risk loci at chromosome 8q24 and identifying *NKX3-1* as a risk gene for prostate cancer. Prostate cancer risk at 8q24 has been well characterized to carry SNPs that are population-specific to African Americans [13, 23, 38]. Previously, up to twelve independent risk signals have been identified at 8q24 in European populations and three significant loci in populations of African ancestries [13, 39] (S2 & S3 Tables).

Our study found four independent clusters of SNPs at this position for the African American study using DAP-G. Two of these four clusters contained SNPs meeting genome-wide significance. Han et al. previously identified rs72725879, rs114798100, and rs111906932 to be three distinct significantly associated loci [13] (S3 Table). All three of these SNPs met genome-wide significance in our study, however, our DAP-G modeling clustered SNPs differently than in the forward selection modeling of Han et al. (Fig 1). In both studies, rs72725879 was identified as an independent risk signal. Han et al. found rs114798100 and rs111906932 to be independent risk loci while DAP-G in our study prioritized rs76595456 as risk signal independent from rs72725879. Interestingly, DAP-G clustered rs114798100 ($PIP = 1.21 \times 10^{-5}$) with rs76595456 ($PIP = .942$), and rs111906932 was not assigned to any cluster by DAP-G. rs76595456 appears to be frequently co-inherited with rs114798100 ($D' = 0.964$ in 1000G AFR) and rs111906932 ($D' = 0.975$ in 1000G AFR). The small number of independent signals identified by DAP-G is unsurprising since DAP-G is a more conservative fine-mapping tool that assumes a single causal variant is expected *a priori*. This difference could also be attributed to sample size since Han et al. had a sample size nearly twice that of our study. Additionally, we found six clusters in the Japanese American study using DAP-G. Of the 102 SNPs assigned to clusters by DAP-G in either African or Japanese American population, only rs72725879 overlapped across populations (Fig 2). rs72725879 has previously been implicated in Asian

ancestry-specific risk to prostate cancer [14], and is in high LD ($r^2$ = 0.815) with rs1456315, which was the most significantly associated SNP in the Japanese American GWAS.

With respect to our identification of *NKX3-1* in the Japanese American TWAS, *NKX3-1* was identified across six tissues including both brain and somatic tissue. *NKX3-1* is a well known tumor suppression gene, whose decreased expression has been associated with prostate cancer [40, 41]. In all six tissues, *NKX3-1* expression was predicted with a negative direction of effect, associating decreased expression with the phenotype. This direction of effect is replicated both in our S-PrediXcan application to the PRACTICAL summary statistics and previous finding in Japanese populations [42].

### rs76595456 is identified as an African ancestry-specific SNP

rs76595456 was the most significantly associated ($P = 1.01 \times 10^{-15}$) SNP in our study. Its minor risk allele is specific to populations of recent African ancestries (MAF = 0.1150), and it is absent in both Asian and European populations [27] (Fig 2). The SNP had an individual PIP of 0.942 and was found in a cluster with seven other SNPs bearing a cluster PIP of 0.995. The SNP is an intron variant of *PCAT2*, a well-established prostate cancer associated transcript [13].

### Novel gene associations implicated by TWAS

In our TWAS of prostate cancer across African Americans, we report *EBPL* as a gene significantly associated (*lfsr* = 0.0423) with prostate cancer in this population. *EBPL* made up the top twenty gene-tissue pairs by *lfsr* (*lfsr* ranging from 0.0423-0.153). It was the most associated gene reported across all five MESA gene expression prediction populations by both P-value ($P = 4.67 \times 10^{-6}$ in AFA) and *lfsr* (*lfsr* = .106 in AFHI). The highest gene-tissue COLOC P4 value, the probability that the GWAS and eQTL signal are colocalized, was 0.18. In the S-PrediXcan analysis of the PRACTICAL data, the gene did not replicate. This failure to replicate could be attributed to difference genetic architecture across the African American test population and the European PRACTICAL population.

The TWAS of prostate cancer in the Japanese American population revealed four unique genes associated with prostate cancer. The previously discussed *NKX3-1* is a well-established tumor suppressor gene whose decreased expression is known to progress the aggressiveness of the tumor [41, 43]. *COQ10B* has not been associated with prostate cancer previously according to the NCBI GWAS Catalog [12]. *PLCL1* has been nominally associated with prostate cancer through a SNP x SNP interaction study [44]. Neither of these genes replicated in the larger S-PrediXcan analysis. *FAM227A* provides an interesting situation since it has been previously identified to be associated with prostate cancer in a GWAS of prostate cancer in Middle Eastern populations [45]. Additionally, it was the only significant gene-tissue pair to be found in prostate tissue. SNPs were not present in the PRACTICAL summary stats to generate a reliably predicted level of expression of *FAM227A* using S-PrediXcan. Despite having nearly 15 million more SNPs in the PRACTICAL summary statistics compared to our GWAS and TWAS of Japanese American populations, SNPs were not genotyped or imputed at this location on chromosome 22 in the PRACTICAL study. One of the SNPs in this model, rs16999186, has a divergent allele frequency across Japanese (MAF = 0.114) and European (MAF = 0.0169) populations [27]. Since this frequency in European populations falls near the quality control MAF threshold in the original PRACTICAL study and below the genotyping threshold for the European-specific designed genotyping array, this SNP could easily be missed in larger studies [8]. *FAM227A* also lies within approximately 150kB of *SUN2* ($P = 1.18 \times 10^{-5}$;*lfsr* = .108) a gene marginally significant in our MESA-HIS prediction model. *SUN2* has recently been identified

as a gene whose decreased expression has been significantly associated with prostate cancer in Japanese populations [46].

## Conclusion

In summary, this study of 4,769 African American and 2,199 Japanese American individuals identifies potential population-specific risk loci for prostate cancer in people of recent African or East Asian ancestries. Since its minor risk allele is found only in populations of recent African ancestries, rs76595456 is suggested as a potentially novel risk SNP to prostate cancer in African Americans. Furthermore, the identification of *FAM227A* as a potential susceptibility gene for prostate cancer in non-European populations highlights growing need for studies of the genetics of prostate cancer in non-European populations. This study's principal limitations were sample size and ancestry matching in gene expression prediction models. Sample sizes of under 5,000 African American subjects and nearly 2,200 Japanese American subjects pale in comparison to studies of exclusively European populations that are nearly two orders of magnitude larger. Considering that African American men are nearly twice as likely to die from prostate cancer as their white counterparts, the need for larger sample sizes of African American subjects cannot be overstated [6]. Regarding the gene expression prediction models used, the GTEx version 8 prediction models are the most comprehensive set of prediction models to date; however, they are built exclusively from the transcriptomes of European ancestries individuals. Where the models provide accuracy and breadth in capturing common eQTLs, they struggle to predict expression from population-specific eQTLs. The MESA prediction models used capture some of the diversity across populations, but they too are limited by sample size (233 African American individuals, 352 Hispanic individuals, and 578 European individuals) and diversity considering there is no model built from transcriptomes of East Asian subjects [29]. To better understand the genetic processes that underlie prostate cancer in diverse populations, more diverse studies are needed.

## Supporting information

**S1 Fig. Quality control PCA against HapMap.** After merging genotypes with those of four reference populations from version three of the HapMap Project, we performed principal component analysis of all three study populations separately. African American (A) and Japanese American (B) genotypes are plotted with three populations from HapMap: Chinese in Beijing and Japanese in Tokyo (ASN), European ancestries in Utah (CEU), and Yoruba people in Ibadan, Nigeria (YRI). The Latin American genotypes are plotted with Chinese in Beijing and Japanese in Tokyo (ASN), European ancestries in Utah (CEU), and indigenous people of North America (NAT).
(TIF)

**S2 Fig. Chromosome 13 GWAS & DAP-G results.** Genome-wide association studies identified no genome-wide significant SNPs. (A) depicts a LocusZoom plots of the most associated GWAS results from Native American population on chromosome 13 [35]. (A) is plotted using 1000G AMR 2014 LD. The y-axis is the -log(P) while the x-axis is location on chromosome 13 measured in megabases. Color represents the LD $r^2$. (B) depicts the results of our GWAS in comparison to DAP-G cluster and PIP for the Latin American population [20]. Each point on the plot represents one SNP in our GWAS. The y-axis is -log(P), and the x-axis is the individual SNP PIP as calculated by DAP-G. The color of each point represents the cluster to which DAP-G assigned it.
(TIF)

**S3 Fig. MESA PrediXcan Manhattan plots.** (A, B, & C) are Manhattan plots of the gene-based association study using MESA monocyte gene expression prediction models for the respective African American, Japanese American, and Latin American populations. Each point represents a gene-tissue test from PrediXcan. The y-axis represents the $-log_{10}(P)$ of the gene-tissue test, and the x-axis plots chromosome number. The size of the dot is inversely proportional to its *lfsr*. (TIF)

**S4 Fig. Latin American PrediXcan Manhattan plots.** Manhattan plot of the gene-based association study using GTEx version 8 *mashr* gene expression prediction models for the Latin American population. Each point represents a gene-tissue test from PrediXcan. The y-axis represents the $-log_{10}(P)$ of the gene-tissue test, and the x-axis plots chromosome number. The size of the dot is inversely proportional to its *lfsr*. (TIF)

**S1 Table. DAP-G clustered SNPs.** At chromosome 8q24, DAP-G calculated PIPs for 12,785 SNPs and 3,878 SNPs in the African American and Japanese American populations, respectively. Of these SNPs with a calculated PIP, 223 SNPs (24 SNPs in African Americans and 199 SNPs in Japanese Americans) were placed into a cluster. Of those placed into a cluster, 102 SNPs at chromosome 8q24 met genome-wide significance in either population. Nine SNPs in the African American study and 93 SNPs in the Japanese American study met genome-wide significance and were placed into a cluster. Of these 102 clustered SNPs meeting genome-wide significance, rs72725879 was the only SNP to overlap across populations. This table contains DAP-G results for the 102 SNPs clustered that met genome-wide significance. (CSV)

**S2 Table. Chromosome 8 SNPs intersecting with NCBI GWAS catalog.** A table that includes all SNPs on chromosome 8 from our African American and Japanese American study that intersect with SNPs from prostate cancer GWAS summary statistics uploaded to the NCBI GWAS Catalog [12]. There are twenty-four published papers with results uploaded to the NCBI GWAS Catalog containing SNPs intersecting with our studies [8, 44, 46–67]. (CSV)

**S3 Table. Comparative results to Han et Al. (2016).** A table comparing the GWAS results from four SNPs on chromosome 8 in our study to the same four SNPs in Han et al. (2016). Two of these SNPs rs76595456 and rs72725879 are prioritized in our study to be independent loci while Han et al. found rs72725879, rs114798100, and rs111906932 as three independent risk loci. BP is the chromosomal location using human genome build 19 [13]. (CSV)

## Acknowledgments

The datasets used for the analyses described in this manuscript were obtained from dbGaP at http://www.ncbi.nlm.nih.gov/projects/gap/cgi-bin/study.cgi?study_id=phs000306.v3.p1.

## Author Contributions

**Conceptualization:** Peter N. Fiorica, Heather E. Wheeler.

**Formal analysis:** Peter N. Fiorica, Ryan Schubert, John D. Morris, Mohammed Abdul Sami.

**Funding acquisition:** Peter N. Fiorica, John D. Morris, Mohammed Abdul Sami, Heather E. Wheeler.

**Methodology:** Peter N. Fiorica, Ryan Schubert, Heather E. Wheeler.

**Project administration:** Heather E. Wheeler.

**Resources:** Peter N. Fiorica, Ryan Schubert, Heather E. Wheeler.

**Supervision:** Heather E. Wheeler.

**Visualization:** Peter N. Fiorica.

**Writing – original draft:** Peter N. Fiorica.

**Writing – review & editing:** Peter N. Fiorica, Ryan Schubert, John D. Morris, Mohammed Abdul Sami, Heather E. Wheeler.

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
