## [Decision Letter · Decision Letter 0]

31 Jul 2020

PONE-D-20-20022

Multi-ethnic transcriptome-wide association study of prostate cancer

PLOS ONE

Dear Dr. Wheeler,

Thank you for submitting your manuscript to PLOS ONE. After careful consideration, we feel that it has merit but does not fully meet PLOS ONE’s publication criteria as it currently stands. Therefore, we invite you to submit a revised version of the manuscript that addresses the points raised during the review process.

1.  The rationale for the study needs to made more clear, particularly in light of previously published work.  What does this study add?  This is especially important as the authors are using some data from previous publications.

2. The authors need to do a more comprehensive literature search on what has already been accomplished in this area (GWAS, eQTLs for risk alleles etc.) and place their study in the context of this previous work.  Details can be added to both the introduction and discussion.

3.  Address reviewer 1's comments on the 8q24 SNP.

4.  Address the other points raised by the reviewers.

We look forward to receiving your revised manuscript.

Kind regards,

Amanda Ewart Toland, Ph.D.

Academic Editor

PLOS ONE

Journal Requirements:

3.We note that [Figure(s) 2] in your submission contain [map/satellite] images which may be copyrighted. All PLOS content is published under the Creative Commons Attribution License (CC BY 4.0), which means that the manuscript, images, and Supporting Information files will be freely available online, and any third party is permitted to access, download, copy, distribute, and use these materials in any way, even commercially, with proper attribution. For these reasons, we cannot publish previously copyrighted maps or satellite images created using proprietary data, such as Google software (Google Maps, Street View, and Earth). For more information, see our copyright guidelines: http://journals.plos.org/plosone/s/licenses-and-copyright.

1.    You may seek permission from the original copyright holder of Figure(s) [2] to publish the content specifically under the CC BY 4.0 license. 

Reviewers' comments:

Reviewer's Responses to Questions

**Comments to the Author**

1. Is the manuscript technically sound, and do the data support the conclusions?

Reviewer #1: Partly

Reviewer #2: Yes

2. Has the statistical analysis been performed appropriately and rigorously? 

Reviewer #1: Yes

Reviewer #2: Yes

3. Have the authors made all data underlying the findings in their manuscript fully available?

Reviewer #1: Yes

Reviewer #2: Yes

4. Is the manuscript presented in an intelligible fashion and written in standard English?

Reviewer #1: Yes

Reviewer #2: Yes

5. Review Comments to the Author

Reviewer #1: The authors utilize available prostate cancer GWAS data to perform GWAS and TWAS of prostate cancer within 3 racial/ethnic groups. The GWAS seems like an academic exercise as the GWAS results for these 3 populations/studies have already been published. A stronger rationale for doing this and re-publishing the results needs to be provided. They also need to acknowledge previous GWAS analyses of these same data as many studies have been published. The results are also not properly framed with regards to known risk variants or regions. For example, the top SNP in the study at 8q24, rs7659456, is correlated with much stronger signals in African Americans which have been published. It is odd that these other stronger SNPs are not found which suggests issues with the imputation. While the TWAS analysis in these studies is novel, the results aren’t particularly interesting as what was found are essentially known eQTL signals with known risk genes.

Minor points:

In abstract and elsewhere, this is not a cohort but rather multiple case-control studies.

The location of the 8q24 SNP is incorrect in Fig 2

Reviewer #2: The study by Fiorica and colleagues is a careful analsyes of genetic susceptibility to prostate cancer across races. This is an impactful area of research and one that is being addressed by various groups. The current findings will therefore resonate with multiple efforts to define the genetic drivers of prostate cancer.

Perhaps, there are technical improvements that could be made (for example, in terms of ancestry inference) but these are minor concerns and the authors justify their approaches within the confines of the study.

6. PLOS authors have the option to publish the peer review history of their article (what does this mean?). If published, this will include your full peer review and any attached files.

Reviewer #1: No

Reviewer #2: **Yes: **Moray J Campbell

---

## [Author Response · Author response to Decision Letter 0]

27 Aug 2020

We thank the editor and reviewers for the thorough examination of our manuscript and for providing positive and helpful feedback. We appreciate the opportunity to address reviewer comments here. Our responses are prefaced by RESPONSE:

Editor’s Comments (Amanda Ewart Toland, Ph.D.):

1. The rationale for the study needs to made more clear, particularly in light of previously published work. What does this study add? This is especially important as the authors are using some data from previous publications.

RESPONSE: 

We thank the editor for highlighting this point and bringing to our attention the need to contextualize our study. In our response to Reviewer 1, we note the novelty of our TWAS and the fact that imputation reference panels have been updated since the time of previous GWAS of this dataset. Our introduction and discussion have been updated:

Lines 68-74

“These GWAS have identified 8q24 as a region carrying susceptibility loci for prostate cancer. While some fine-mapping of this region has been done in diverse populations, transcriptome-wide association studies (TWAS) of prostate cancer have not been conducted across diverse populations [13, 14]. TWAS serves as a systematic method for integrating expression quantitative trait loci (eQTL) data from GWAS [15-17]. Since TWAS uses gene expression as an intermediate phenotype, it has a functional advantage over GWAS.”

Lines 246-250

“Broadly, the findings of this study confirm previously well-established information about the genetics of prostate cancer in diverse populations; however, our study differs from previous ones since we performed the first TWAS of prostate cancer in these populations using mashr prediction models from GTEx version 8 and diverse prediction models from MESA [17, 29].”

2. The authors need to do a more comprehensive literature search on what has already been accomplished in this area (GWAS, eQTLs for risk alleles etc.) and place their study in the context of this previous work. Details can be added to both the introduction and discussion.

RESPONSE:

We appreciate this comment from the editor. After performing a fuller literature review, we have contextualized our study primarily with respect to Han et al. (2016) JNCI. Supplemental Table 3 details the four SNPs we explore in our discussion. We discuss the differences in our findings and potential reasons for these differences in our discussion when we state:

Lines 253-269

“Previously, up to twelve independent risk signals have been identified at 8q24 in European populations and three significant loci in populations of African ancestries [13,38] (Supplemental Table 2).

Our study found four independent clusters of SNPs at this position for the African American study using DAP-G. Two of these four clusters contained SNPs meeting genome-wide significance. Han et al. previously identified rs72725879, rs114798100, and rs111906932 to be three distinct significantly associated loci [13](Supplemental Table 3). All three of these SNPs met genome-wide significance in our study, however, DAP-G clustered SNPs differently than their findings (Fig 1) . In both studies, rs72725879 was identified as an independent risk signal. Han et al. found rs114798100 and rs111906932 to be independent risk loci while DAP-G in our study prioritized rs76595456 as risk signal independent from rs72725879. Interestingly, DAP-G clustered rs114798100 (PIP=1.21x10-5) with rs76595456 (PIP=.942), and rs111906932 was not assigned to any cluster by DAP-G. rs76595456 appears to be frequently co-inherited with rs114798100 (D'=0.964 in 1000G AFR) and rs111906932 (D'=0.975 in 1000G AFR). The small number of independent signals identified by DAP-G is unsurprising since DAP-G is a more conservative fine-mapping tool that assumes a single causal variant is expected a priori.”

3. Address reviewer 1's comments on the 8q24 SNP.

RESPONSE:

We have since relabelled Figure 2 appropriately. We further explore literature regarding the 8q24 region in African American prostate cancer in our discussion and address this point in our response to Reviewer 1. 

4. Address the other points raised by the reviewers.

RESPONSE:

We appreciate the careful and thorough review of our manuscript by the reviewers. We have outlined our response to reviewers below.

We note that you have stated that you will provide repository information for your data at acceptance. Should your manuscript be accepted for publication, we will hold it until you provide the relevant accession numbers or DOIs necessary to access your data. If you wish to make changes to your Data Availability statement, please describe these changes in your cover letter and we will update your Data Availability statement to reflect the information you provide.

RESPONSE:

No data availability changes have been made. All scripts, notes, and results from this study can be found at https://github.com/WheelerLab/Prostate-Cancer-Study. We have also submitted our GWAS summary statistics to the NCBI GWAS Catalog for our GWAS in all three populations.

We note that [Figure(s) 2] in your submission contain [map/satellite] images which may be copyrighted. All PLOS content is published under the Creative Commons Attribution License (CC BY 4.0), which means that the manuscript, images, and Supporting Information files will be freely available online, and any third party is permitted to access, download, copy, distribute, and use these materials in any way, even commercially, with proper attribution. For these reasons, we cannot publish previously copyrighted maps or satellite images created using proprietary data, such as Google software (Google Maps, Street View, and Earth). For more information, see our copyright guidelines: http://journals.plos.org/plosone/s/licenses-and-copyright.

RESPONSE:

The Geography of Genetic Variants Browser, the source from which Figure 2 was adapted, has been made public by Creative Commons Attribution License (CC BY 4.0).

Joseph H Marcus, John Novembre, Visualizing the geography of genetic variants, Bioinformatics, Volume 33, Issue 4, 15 February 2017, Pages 594–595, https://doi.org/10.1093/bioinformatics/btw643

Comments to the Author

5. Review Comments to the Author

Reviewer #1: The authors utilize available prostate cancer GWAS data to perform GWAS and TWAS of prostate cancer within 3 racial/ethnic groups. The GWAS seems like an academic exercise as the GWAS results for these 3 populations/studies have already been published. A stronger rationale for doing this and re-publishing the results needs to be provided. 

RESPONSE:

We appreciate Reviewer 1’s comment to provide a stronger rationale for the study. We have addressed this at:

Lines 66-83

“Of the 67 published prostate cancer GWAS in the National Center for Biotechnology Information (NCBI) GWAS Catalog, only 16 studies included individuals of non-European ancestries [12]. These GWAS have identified 8q24 as a region carrying susceptibility loci for prostate cancer. While some fine-mapping of this region has been done in diverse populations, transcriptome-wide association studies (TWAS) of prostate cancer have not been conducted across diverse populations [13,14]. TWAS serves as a systematic method for integrating expression quantitative trait loci (eQTL) data from GWAS [15-17]. Since TWAS uses gene-expression as an intermediate phenotype, it has a functional advantage over GWAS. One of the largest GWAS published up to this point that included over 140,000 individuals of European ancestries was the Prostate Cancer Association Group to investigate Cancer-Associated Alterations in the Genome (PRACTICAL) Consortium [8]. Two of the largest gene-based association studies specific to prostate cancer were TWAS of the PRACTICAL GWAS summary statistics [18,19]. While these studies provided insight into genes associated with prostate cancer in European subjects, they provided little insight into genes affected by ancestry-specific SNPs in diverse populations. Moreover, broader and more accurate genotype and gene expression imputation panels have been created since the studies in diverse populations were published.”

Lines 85-90

“To do this, we performed a standard case-control GWAS across 4,769 African American subjects, 2,147 Latin American subjects, and 2,199 Japanese American subjects. We used a deterministic approximation of posteriors for GWAS (DAP-G) to fine-map the 8q24 susceptibility region [20]. In addition to GWAS, we performed TWAS using PrediXcan across 46 tissues from GTEx version 8 and five models from the Multi-Ethnic Study of Atherosclerosis. We replicated our data in an S-PrediXcan application to the PRACTICAL summary statistics [15, 21].” 

Lines 246-250

“Broadly, the findings of this study confirm previously well-established information about the genetics of prostate cancer in diverse populations; however, our study differs from previous ones since we performed the first TWAS of prostate cancer in these populations using mashr prediction models from GTEx version 8 and diverse prediction models from MESA [17, 29].”

They also need to acknowledge previous GWAS analyses of these same data as many studies have been published.

RESPONSE:

We thank Reviewer 1 for helping us recognize the need to contextualize our results. We have included Supplemental Table 3 that includes results of twenty-six different GWAS included in the NCBI GWAS Catalog that contain SNPs that intersect with SNPs in our studies at chromosome 8. The table includes P-values from both our study and the reference GWAS Catalog study. Additionally, it includes sample sizes and ancestries for each study. It should be noted that Han et al. (2016) JNCI, which we discuss in our next comment, is not included in the NCBI GWAS Catalog nor the citations for this set of genotypes and phenotypes in dbGaP. This explains a potential reason that this study was overlooked in our initial literature review.

 The results are also not properly framed with regards to known risk variants or regions. For example, the top SNP in the study at 8q24, y, is correlated with much stronger signals in African Americans which have been published. It is odd that these other stronger SNPs are not found which suggests issues with the imputation. 

RESPONSE:

Thank you for this suggestion. We recognize that rs72725879 and rs76595456 have previously been identified with more significance in Han et al. (2016) JNCI (N=9,531). We attribute this difference to the fact that the Han et al. study has nearly double the sample size in both their discovery and replication populations. Below is a table comparing the P-values of the Han et al. study and our African American GWAS.

SNP. P-Value (Han et al.) n=9531. P-Value (Fiorica et al.) n=4769

rs72725879. 1.07E-23. 3.68E-12

rs76595456. 1.75E-32. 1.01E-15

rs114798100. 1.61E-33 3.38E-14

rs111906932. 4.32E-10. 2.46E-09

We have further explained this in the paper at lines 253-276, where we write:

“Previously, up to twelve independent risk signals have been identified at 8q24 in European populations and three significant loci in populations of African ancestries [13, 38] (S2 Table and S3 Table).

Our study found four independent clusters of SNPs at this position for the African American study using DAP-G. Two of these four clusters contained SNPs meeting genome-wide significance. Han et al. previously identified rs72725879, rs114798100, and rs111906932 to be three distinct significantly associated loci [13] (S3 Table). All three of these SNPs met genome-wide significance in our study, however, our DAP-G modeling clustered SNPs differently than in the forward selection modeling of Han et al. (Fig 1). In both studies, rs72725879 was identified as an independent risk signal. Han et al. found rs114798100 and rs111906932 to be independent risk loci while DAP-G in our study prioritized rs76595456 as risk signal independent from rs72725879. Interestingly, DAP-G clustered rs114798100 (PIP=1.21x10-5) with rs76595456 (PIP=.942), and rs111906932 was not assigned to any cluster by DAP-G. rs76595456 appears to be frequently co-inherited with rs114798100 (D'=0.964 in 1000G AFR) and rs111906932 (D'=0.975 in 1000G AFR). The small number of independent signals identified by DAP-G is unsurprising since DAP-G is a more conservative fine-mapping tool that assumes a single causal variant is expected a priori. Additionally, we found six clusters in the Japanese American study using DAP-G. Of the 102 SNPs assigned to clusters by DAP-G in either African or Japanese American population, only rs72725879 overlapped across populations (Fig 2). rs72725879 has previously been implicated in Asian ancestry-specific risk to prostate cancer [14], and is in high LD (r2=0.815) with rs1456315, which was the most significantly associated SNP in the Japanese American GWAS.”

In addition, we now include a new supplemental table (S3 Table) containing overlap between our results, and those of the GWAS catalog. Of the 12 different prostate cancer GWAS we found with intersecting genome-wide significant SNPs, seven of the studies contained SNPs with P-values more significant than those identified in our analysis. We attribute this discrepancy to a combination of weaker statistical power due to lower sample size and more stringent imputation filtering in our study. Below are details further explaining these differences in significance:

Hoffman TJ et al. (2015), which accounts for four of the fifteen SNPs with different P-values used r2 ≥ 0.3 (compared to our r2 ≥ 0.8), so there may have been less accurately imputed SNPs in their analysis, potentially inflating P-values. Moreover, their sample included nearly 39,000 non-Hispanic white subjects.

Cheng I et al. (2012), Takata R et al. (2010), and Takata R et al. (2019) all used r2 ≥ 0.3 for their imputation filtering.

Gudmundsson J et al. (2009) and Gudmundsson J et al. (2007) perform their analyses in samples of over 37,000 and 4,500 European individuals, respectively.

Knipe DW et al. (2014) and Schumacher FR et at. (2011) performed their analyses in exclusively European populations.

While the TWAS analysis in these studies is novel, the results aren’t particularly interesting as what was found are essentially known eQTL signals with known risk genes.

RESPONSE:

We thank the Reviewer for recognizing the novelty of our TWAS analysis. We acknowledge that NKX3-1 is a well-characterized prostate cancer risk gene regulated by an enhancer at the 8q24 region that replicated in our S-PrediXcan application of the PRACTICAL summary statistics. Besides this gene, the other four genes we discovered (Table 2) are not located on chromosome 8 and have not been implicated as eQTLs in prostate cancer previously. While these genes did not replicate, there is potentially new biology to be explored in this area in non-European populations. The two largest prostate cancer TWAS to date, Mancuso et al. (2018) Nat Com and Wu et al. (2019) Can Res, use GTEx version 6 prediction models. We note that while these findings might be false positives, we used prediction models that are both more diverse and more novel. We note this at lines 246-250, where we state:

“Broadly, the findings of this study confirm previously well-established information about the genetics of prostate cancer in diverse populations; however, our study differs from previous ones since we performed the first TWAS of prostate cancer in these populations using mashr prediction models from GTEx version 8 and diverse prediction models from MESA [13, 38].”

Minor points:

In abstract and elsewhere, this is not a cohort but rather multiple case-control studies.

RESPONSE:

We appreciate the Reviewer bringing this point to our attention. We have changed the language surrounding each case-control study to reflect this.

The location of the 8q24 SNP is incorrect in Fig 2

RESPONSE: 

We apologize for this mistake. The SNP in Figure 2A was mislabeled as rs7659456 when the SNP in our study was rs76595456. The position for rs76595456 was correctly labeled, but there was a missing number in the SNP ID. We have corrected any mislabeled SNPs and changed the language in the Figure 2 caption to read: 

“A depiction of the global minor allele frequencies rs76595456 (A), rs72725879 (B), and rs1456315 (C). (A) rs76595456 represents the most significantly associated SNP in the African American GWAS. The minor allele, T, is found only in populations of recent African ancestries. (B) rs72725879 represents the only SNP to be identified by DAP-G in a cluster in both the African American and Japanese American GWAS. (C) rs1456315 represents the most significantly associated SNP in the Japanese American GWAS. rs1456315 (C) is found in strong LD with rs72725879 (B) (r2=0.815) in the Japanese American GWAS population. rs1456315 and rs72725879 are not linked in the African American GWAS population (r2=0.448). This figure was adapted from one generated using the Geography of Genetic Variants Browser [36]. SNP position on chromosome 8 is labeled using hg19 coordinates from 1000G.” 

Reviewer #2: The study by Fiorica and colleagues is a careful analsyes of genetic susceptibility to prostate cancer across races. This is an impactful area of research and one that is being addressed by various groups. The current findings will therefore resonate with multiple efforts to define the genetic drivers of prostate cancer.

Perhaps, there are technical improvements that could be made (for example, in terms of ancestry inference) but these are minor concerns and the authors justify their approaches within the confines of the study.

RESPONSE: We thank Reviewer 2 for his careful review and appreciation of our work. We acknowledge his comment regarding technical improvements with respect for ancestry inference. We agree that these would be excellent improvements to pursue in future studies with larger sample sizes.

---

## [Decision Letter · Decision Letter 1]

11 Sep 2020

Multi-ethnic transcriptome-wide association study of prostate cancer

PONE-D-20-20022R1

Dear Dr. Wheeler,

We’re pleased to inform you that your manuscript has been judged scientifically suitable for publication and will be formally accepted for publication once it meets all outstanding technical requirements.

Kind regards,

Amanda Ewart Toland, Ph.D.

Academic Editor

PLOS ONE

Additional Editor Comments (optional):

Reviewers' comments:

Reviewer's Responses to Questions

**Comments to the Author**

1. If the authors have adequately addressed your comments raised in a previous round of review and you feel that this manuscript is now acceptable for publication, you may indicate that here to bypass the “Comments to the Author” section, enter your conflict of interest statement in the “Confidential to Editor” section, and submit your "Accept" recommendation.

Reviewer #1: All comments have been addressed

Reviewer #2: All comments have been addressed

2. Is the manuscript technically sound, and do the data support the conclusions?

Reviewer #1: Yes

Reviewer #2: Yes

3. Has the statistical analysis been performed appropriately and rigorously? 

Reviewer #1: Yes

Reviewer #2: Yes

4. Have the authors made all data underlying the findings in their manuscript fully available?

Reviewer #1: Yes

Reviewer #2: Yes

5. Is the manuscript presented in an intelligible fashion and written in standard English?

Reviewer #1: Yes

Reviewer #2: Yes

6. Review Comments to the Author

Reviewer #1: (No Response)

Reviewer #2: thoughtful comments to generally positive reviews. The comments by the reviewers highlighted certain issues which have been answered

7. PLOS authors have the option to publish the peer review history of their article (what does this mean?). If published, this will include your full peer review and any attached files.

Reviewer #1: No

Reviewer #2: **Yes: **Moray J Campbell

---

## [Editor Report · Acceptance letter]

17 Sep 2020

PONE-D-20-20022R1 

Multi-ethnic transcriptome-wide association study of prostate cancer  

Dear Dr. Wheeler:

I'm pleased to inform you that your manuscript has been deemed suitable for publication in PLOS ONE. Congratulations! Your manuscript is now with our production department. 

Kind regards, 

on behalf of

Dr. Amanda Ewart Toland 

Academic Editor

PLOS ONE